# Precise Filtration of Chronic Myeloid Leukemia Cells by an Ultrathin Microporous Membrane with Backflushing to Minimize Fouling

**DOI:** 10.3390/membranes13080707

**Published:** 2023-07-29

**Authors:** Jaehyuk Lee, Jeongpyo Hong, Jungwon Lee, Changgyu Lee, Tony Kim, Young Jeong, Kwanghee Kim, Inhwa Jung

**Affiliations:** 1Department of Mechanical Engineering, Kyung Hee University, Yongin 17104, Republic of Korea; jh.lee@metapore.com; 2R&D Center, Metapore Co., Ltd., Advanced Institutes of Convergence Technology 8F, Suwon 16229, Republic of Korea; hongjp05@metapore.com (J.H.); jyl9965@metapore.com (J.L.); cglee@metapore.com (C.L.); tonykim@metapore.com (T.K.); young.j@metapore.com (Y.J.); 3National NanoFab Center, 291 Daehak-ro, Yuseong-gu, Daejeon 34141, Republic of Korea; khkim@nnfc.re.kr

**Keywords:** ultra-thin porous membrane, pumping head filtration with backflush, chronic myeloid leukemia cell, fouling, blocking filtration model

## Abstract

A cell filtration platform that affords accurate size separation and minimizes fouling was developed. The platform features an ultra-thin porous membrane (UTM) filter, a pumping head filtration with backflush (PHF), and cell size measurement (CSM) software. The UTM chip is an ultrathin free-standing membrane with a large window area of 0.68 mm^2^, a pore diameter of 5 to 9 μm, and a thickness of less than 0.9 μm. The PHF prevents filter fouling. The CSM software analyzes the size distributions of the supernatants and subnatants of isolated cells and presents the data visually. The *D*99 particle size of cells of the chronic myeloid leukemia (CML) line K562 decreased from 22.2 to 17.5 μm after passage through a 5-μm filter. K562 cells could be separated by careful selection of the pore size; the recovery rate attained 91.3%. The method was compared to conventional blocking models by evaluating the mean square errors (MSEs) between the measured and calculated filtering volumes. The filtering rate was fitted by a linear regression model with a significance that exceeded 0.99 based on the *R*^2^ value. The platform can be used to separate various soft biomaterials and afford excellent stability during filtration.

## 1. Introduction

Biomaterial filtration is used to remove unwanted substances or to isolate materials required for disease diagnoses. Such filtration is of great commercial importance in the field of clinical pathology. Typically, materials such as red blood cells (RBCs), white blood cells (WBCs), and circulating tumor cells (CTCs) are filtered. Various methods have been developed to isolate such microscopic materials [1]. The most common methods feature antigen-antibody reactions or employ microfluidics-based devices to isolate minute microvesicles (MVs) or cells [2,3]. As the method applying antigen-antibody reactions shows high separation efficiency, it has been developed in various ways. In methods that use antibodies on magnetic beads, the sizes of CTCs are artificially amplified by binding antibody-attached beads to the cell surfaces, facilitating cell isolation [4,5]. One method isolates CTCs using a microvortex generated by a herringbone structure to deliver cells to an antibody-rich region [6]. Another method captures CTCs using an antibody-attached OncoBean Chip [7]. However, methods that employ antibodies require inconvenient post-processing to retrieve the antigens [8]. Pinched flow fractionation exploits only the flow control afforded by microfluidic devices; this minimizes contact between substances of interest and device structures [9,10]. A method using an acoustic node generated by sound waves has been employed to isolate smaller particles using a microfluidic device. Although particles smaller than 1 μm could thus be isolated, the separation efficiency fell when the size difference between particles was not high [11,12].

Membrane filtration is a physical method that isolates particles without using specific biomarkers such as antibodies; also, sophisticated microfluidic devices and flow control are not required. Particles of various sizes are isolated by adjusting the membrane pore size, density, and thickness. If two membrane filters are employed, the size ranges of filtered particles can be controlled. One method for isolation of MVs sized 20–600 nm employed two polycarbonate track etched filters [13]. Anodizing aluminum oxide (AAO) filters effectively isolate nanoparticles sized 24–150 nm [14]. Such filters are simple to use, but the thick membranes are readily foul, and the low pore density and poor size uniformity prevent accurate size-based separation [15]. More accurate separation is possible when the filtering device is fabricated using microelectromechanical systems (MEMS) technologies that afford high pattern uniformity. Deterministic lateral displacement pillar arrays fabricated via MEMS generate fluid bifurcations within structures, and effectively isolate WBCs from whole blood [16]. A modification of the technique has been used to isolate exosomes sized 20–110 nm [17]. Another method isolated WBCs and extracted the RNAs therein [18]. However, fouling was of concern because the material being filtered progressed through a long, rod-embossed patterned passage.

Fouling and inaccurate size separation can be significantly improved using highly porous ultrathin membranes (UTMs) prepared via MEMS. This is a very simple, 2D, physical, particle isolation mechanism. In addition to the commonly used silicon-based thin films, polymer-based porous membranes can also be employed [19]. Photolithography is usually employed to pattern the pore array, although e-beam lithography affords more precise patterning. Recently, a new lithography technique featuring gaps between uniformly arranged beads has been reported [20]. Silicon and the oxides (SiO_2_) and nitrides (Si_3_N_4_) thereof are the principal filter materials, and are non-toxic, allow cell culture, are very biocompatible, and do not cause cell aggregation or damage [21].

During filtration, fouling can be quantified by the recovery rate. If that rate is high, losses are reduced, improving filtering performance and enabling long-term filter use. Although an excellent filter is required to ensure a high recovery rate, the filtration method is also important. In general, dead end filtration (DEF) is a unidirectional filtration method that easily blocks filters, whereas tangential flow filtration (TFF) prevents blocking and increases filtration continuity [22]. However, even during DEF, the recovery rate can be increased by backflushing [23]. Also, a high recovery rate can be ensured using well-established operating protocols featuring multiple filters and buffers [24]. 

Here, we develop a micro-scale cell filtering platform that isolates certain types of cells more accurately than other types. The platform features a UTM fabricated via MEMS and a pumping head filtration with backflush (PHF). Cell permeability is increased by the UTM, and the recovery rate is maximized by the PHF. Preliminary testing employed polystyrene (PS) beads, and verification used the chronic myeloid leukemia (CML) cell line K562. As the particle size distribution was broad, it was possible to evaluate both size separation and the recovery rate [25]. To verify the effects of backflush, the filtering rates were compared to those of four blocking filtration laws, and the validity of the comparisons was discussed [26]. Typically, ImageJ software is used to measure cell size [27], but the software does not evaluate multiple cells; we thus developed cell size measurement (CSM) software that sized large numbers of cells. The CSM uses deep learning to identify live cells and generates histograms of cell size distributions, enabling rapid statistical analyses of mean cell sizes and the standard deviations. The cell filtration platform featuring the UTM, PHF, and CSM software allowed CTC and CML cell filtration and will find many commercial diagnostic applications.

## 2. Method

### 2.1. Fabrication Processes of Ultra-Thin Porous Membrane (UTM) Filter

The fabrication processes of UTM are shown in Figure 1a. A SiO_2_ thin film of 5000 Å was formed on the front side of a silicon wafer of orientation <100> with an oxidation furnace (Centronic E1200, Centrotherm, Blaubeuren, Germany). Afterward, a Si_3_N_4_ thin film of 9000 Å was deposited on top of the film by low pressure chemical vapor deposition (LP-CVD) method. The pore array was patterned by photolithography (I-Line Stepper NSR2205i12D, Nikon, Tokyo, Japan), and only the Si_3_N_4_ layer was etched by a dry etching process (ICP dry etcher PlasmaPro100 Cobra, Oxford, UK). For the etching, the wafer was exposed to plasma by CHF_3_ gas for 13 min. On the backside of the wafer, the membrane area was patterned in a square shape by photolithography (Mask-Aligner MA200 Compact, SUSS Micro Tec, Garching, Germany). Then, the double dielectric layer was over etched until the Si was exposed. To form a membrane by etching Si, wet etching was performed by exposing the wafer to a 20% KOH solution at 90 °C for 3 h (Potassium hydroxide solution 45%, Daejung Chemical Co., Siheung-si, Korea). To etch the residual SiO_2_ film below the pore patterned Si_3_N_4_ film, the wafer was soaked in the BOE etchant for 7 min (Buffered oxide etch 6:1, Samchun Chemical, Pyeongtaek-si, Republic of Korea).

### 2.2. Operation of the Filtering System with Backflush

The structure of PHF and the operation principle of the backflush are shown in Figure 1b. The filtering system consists of a UTM filter, a pump head, a filter housing, and a syringe pump (Fusion 100, Chemyx, Stafford, TX, USA). To control the step motor in the syringe pump, a Linux-based 8-bit MCU embedded system (ATmega 328) was used. A GUI program was developed by Python for driving the embedded system. The reciprocating piston motion of the syringe pump was applied to continuously disperse the cells to prevent fouling of the filter. The detailed method of back flushing in the filtering process is as follows. In the upper part of PHF, 3 ports are formed on the top of it, an open/close port on the left, an injection/suction port in the middle, and a buffer injection port on the right. PHF operates in two steps. In step 1, positive pressure is applied to the middle port to proceed with filtering. At this time, the left port is opened by the solenoid valve. DPBS buffer solution is continuously supplied to the right port at a flow rate of 1 to 4 mL/min depending on the filtration rate. In step 2, negative pressure is applied to the middle port to prevent fouling of the membrane filter. At this time, the left port should be closed to ensure a complete seal. To optimize filtering, steps 1 and 2 of the protocol operated for 5 s each, and were then repeated. The PHF system is similar to the existing systems, but is much more compact and efficient [23]. For TFF, two syringe pumps (PSD/4 Syringe Pump Drive, Hamilton, Reno, NV, USA) were utilized and controlled for continuous liquid injection. (Further details on the DEF, TFF, and PHF systems are shown in Appendix A.)

### 2.3. Filtering Rate Measurements and Fitting to the Blocking Laws

As shown in the lower panel of Figure 1b, the filtering rate was measured by configuring a bench-scale microfiltration system. The filtering rate was used to verify the validity of the blocking model [28]. Three filtration methods, thus DEF TFF, and backflush filtration were evaluated. The backflush filtration was applied by the PHF system. To determine the filtering rate, the weight of the filtrate was measured using a data acquisition program (RsWeight, Ver5.4) of an electronic micro balance (HR-200, A&D Co., Tokyo, Japan). The collected data were further analyzed by fitting with blocking models i.e., complete blocking, intermediate blocking, standard blocking, and cake filtration law.

### 2.4. Filtration Processes of Polystyrene (PS) Beads and CML (K562) Cells

To filter PS beads, four types of beads with diameters of 5.12, 6.78, 8.91 and 10.6 μm (Uniform Polystyrene Latex, Magphere Inc., Goshen, IN, USA) were dispersed in DI water in equal numbers. A solution of 10 mL with a particle density of about 1 × 10^6^ EA/mL was prepared. The solution was filtered for 30 min using a UTM filter with a pore diameter of 6, 7, 8, and 9 μm. For analysis, filtrate and retentate were concentrated using a centrifuge (Centrifuge 5910 R, Eppendorf, Germany). 10 mL of filtrate and retentate were concentrated to 1 mL, respectively. Similarly, in the case of K562 cells, 10 mL of solution with a particle concentration of about 1 × 10^6^ EA/mL was prepared. The average size of K562 cells in the original solution before filtration was 15.9 μm. The K562 cell solution was filtered for 40 min using a UTM filter with a pore diameter of 6, 7, 8, and 9 μm followed by centrifugation. 

### 2.5. Determination of Size Distribution and the Recovery Rate

The distribution of the cell sizes was determined to evaluate the separation performance. By analyzing optical microscope images of the cells, the size distribution was determined. To prepare samples for the optical microscope (Eclipse E400, Nikon, Japan), the concentrated solutions of K562 in filtrate, retentate, and the original solution were mixed with a cell staining reagent (Trypan Blue, Invitrogen, Waltham, MA, USA) at a 1:1 ratio, and then 10 μL of the mixture was injected into a cell counting slide (Cell Counting Chamber Slides, Invitrogen, USA). The cells inside the slide were stabilized for 1 min. Then, 100× magnification images were acquired at least 15 points. The CSM software converts the image into gray scale, filters out noise and agglomerated objects. By applying a deep learning algorithm, dead cells were recognized and excluded. Finally, the total area and average diameter were determined from the lines formed at the cell edges. From the size distribution histogram of the cells, the values of *D*10, *D*50, *D*90, *D*95 and *D*99 were determined. (Further details on the recognition of cells and size determination are shown in Appendix A).

To determine the recovery rate, the concentrated K562 cell solutions were analyzed with a cell counter (Countess II, Invitrogen, USA). Three types of solutions were mixed with the staining reagent and injected into the cell counting slides. To obtain the number of particles, at least three samples were used. The total recovery rate was determined by the following equation:(1)$$total\;recovery\;rate\;(\%)=\frac{N_{filtrate}+N_{retentate}}{N_{original}}\times 100$$
where, *N*_filtrate_ is the number of particles in the filtrate, *N*_retentate_ is the value for the retentate, and *N*_original_ is the value for the original solution. The recovery rate in the filtrate was determined by *N*_filtrate_/*N*_original_ and the recovery rate in the retentate was determined by *N*_retentate_/*N*_original_.

### 2.6. Observation of Structural Characteristics of UTM Filters

To determine structural characteristics, UTM filters were observed using FE-SEM (Helios 5 UC DualBeam, FEI, Hillsboro, OR, USA). The top view and tilt view of UTM filters were acquired. For low magnification, 1300× magnification images were acquired and 10,000× magnification images were acquired for high magnification. To measure the film thickness, the device was etched using Focused Ion Beam (FIB) and measured at 10,000× magnification. 

### 2.7. Procedure of Fluorescence Microscopy Observation of K562 Cells

For culturing K562 cell, about 1.0 × 10^6^ cells were thawed in a T25 flask. Cell passaging was conducted at 2-day intervals with a cell seeding condition of 2.0 × 10^5^ cells/mL. For cell culture media, Iscove Modified Dulbecco Medium (IMDM, Gibco, Waltham, MA, USA), HyClone Characterized Fetal Bovine Serum (FBS, Gibco) 10%, and penicillin/streptomycin 1% were used. Subculturing was performed after harvesting 8.0 × 10^5^ cells/mL. The third passage sample was used for the experiment. Incubation was conducted in an incubator (ICO240, Memmert, Schwabach, Germany) at 37 °C and an atmosphere condition of 5% CO_2_. 

For cell staining, a fluorescence in situ hybridization (FISH) probe (Vysis LSI BCR/ABL Dual Color, Dual Fusion Translocation probe kit, 08L10-001, Gibco, USA) was used. Cells were stained in the following processes. After suspending the cells for staining, a drop (10 μL) was deposited on a pre-washed slide glass and then dried. Afterwards, the slide glass was immersed in 2× SSC buffer for 2 min. Then, the slide glass was immersed in 70%, 85%, and 100% ethanol solutions for 2 min each, then dehydrated and dried. For the hybridization of K562 cells, care was taken to avoid exposing the cells to light. The probe staining solution was pre-warmed at 37 °C in a heat block for 5 min. The pre-warmed solution was deposited on the cells immobilized on a slide glass. Then, the area was slowly covered with a coverslip. The coverslip was tilted 45° to avoid air bubbles. For sealing, the sample was coated with a rubber solution and reacted at 75 °C for 2 min on a thermal stage. Finally, the sample was placed in a light-proof container and reacted under conditions of humidity of 95% to 100%, temperature of 37 °C, and reaction time of 9 to 18 h. After hybridization, the coverslip of the sample is removed and immersed in a 0.4× SSC + 0.3% NP-40 buffer at 72 °C for 2 min for washing. Then, the sample was immersed in a 2× SSC + 0.1% NP-40 buffer at room temperature for 30 s. 

To observe with a fluorescence microscope (Optiphot-2 Fluorescence Microscope, Nikon, Japan), DAPI staining solution is deposited for nuclear staining after removing surface moisture. Then, the area was slowly covered with a coverslip. After stabilization in a dark area for 10 min, the sample was used for the observation.

### 2.8. Finite Element Analysis (FEA)

Finite element analysis was performed using ANSYS R16 software. The cells were modeled as spheres of diameter 15 μm, and the device material was assumed to be viscoelastic. For Young’s modulus and Poisson’s ratio, values reported for K562 cells were applied [29,30]. The viscoelastic properties were modeled as a prony relaxation series using the viscoelasticity of a typical biological tissue [31]. (More details on the viscoelastic property of the cell can be found in Appendix A). The thickness of the UTM filter was 0.9 μm, and three pore diameters of 5, 6, and 9 μm were modeled. The material was assumed to be isotropically elastic, and the Young modulus and Poisson ratio were those of Si_3_N_4_ [32]. The contacts between cells and filters were assumed to be either bonded or frictional. In the latter case, the friction coefficient was assumed to be 0.3. The range of convergence of the solutions tended to decrease when friction was applied. Although the difference between the results with and without friction was not large, a recent report showed that this difference clearly increased as the pressure difference increased [33].

## 3. Theory

To explain filter fouling in the filtration process, a number of theoretical models have been proposed. In particular, Ruth’s cake filtration model [34,35] and Hermans and Bredee’s blocking model [36] are recognized as the earliest significant achievements. Later, the models were established as four blocking filtration laws by Grace [37,38] and Hermia [39] and other researchers. As shown in Figure 1c, the blocking filtration laws consist of cake filtration, intermediate blocking, standard blocking, and complete blocking. In the cake filtration model, aggregation between particles is the main cause of fouling, while in the complete blocking model, the blocking of pores by the particles is considered to determine fouling. In the intermediate blocking and standard blocking models, there are aspects that include both mechanisms.

According to the blocking filtration law, the filtration under constant pressure can be expressed as a common differential equation [26]:(2)$$\frac{d^{2}t}{d\nu^{2}}={k\left(\frac{dt}{d\nu }\right)}^{n}$$
where, *v* is the filtrate volume per unit membrane area (*m*), and *n* is the blocking index which is different for each filtration model, where 0 is the value for cake filtration, 1 for intermediate blocking, 1.5 for standard blocking, and 2 for complete blocking. While *n* can be simply determined according to each blocking model, the blocking constant *k* is related to various filtering conditions, which is difficult to determine. 

For cake filtration (*n* = 0), Equation (2) is derived as follows:(3)$$\nu (t)=\frac{\sqrt{1+2k_{c}J_{0}^{2}t}-1}{{k_{c}J}_{0}}$$
(4)$$J(t)=\frac{d\nu }{dt}=\frac{J_{0}}{\sqrt{1+2k_{c}J_{0}^{2}t}}$$
where, *k*_c_ is the blocking constant for the cake filtration model (m^−2^s), *J* is the filtration rate (m/s), and *J*_0_ is the initial filtration rate.

For the intermediate blocking model (*n* = 1), Equation (2) is derived as follows:(5)$$\nu (t)=\frac{1}{k_{i}}\ln\left(1+k_{i}J_{0}t\right)$$
(6)$$J(t)=\frac{J_{0}}{1+k_{i}J_{0}t}$$
where, *k*_i_ is the blocking constant for the intermediate blocking model (m^−1^). 

For the standard blocking model (*n* = 1.5), Equation (2) is derived as follows:(7)$$\nu (t)=\frac{1}{\frac{k_{s}}{2}+{\left(J_{0}t\right)}^{-1}}$$
(8)$$J(t)=\frac{J_{0}}{{\left(1+\frac{k_{s}}{2}J_{0}t\right)}^{2}}$$
where, *k*_s_ is the blocking constant for the standard blocking model (m^−1^).

For the complete blocking model (*n* = 2), Equation (2) is derived as follows:(9)$$\nu (t)=\frac{J_{0}}{k_{b}}\left(1-e^{-k_{b}t}\right)$$
(10)$$J(t)=J_{0}e^{-k_{b}t}$$
where, *k*_b_ is the blocking constant for the complete blocking model (s^−1^).

Research on verifying the validity of four models for each filtering method was conducted. A MATLAB program was created to realize a routine that minimizes the error between the measured filtrate volume and the calculated value. Hereby, the validity of each modeling according to the filtering method was verified, and the blocking constant could be obtained.

## 4. Results and Discussion

Photographs of the UTM filter are shown in Figure 2a. Si_3_N_4_ membranes are arranged in a rectangular array of 10 columns and 13 rows on a 25 × 25 mm silicon substrate. The hexagonal pore arrays are of diameters (*d*_pore_) 5, 6, 7, 8, and 9 μm and 0.9 μm in thickness (*t*_pore_). The distance between pores was chosen to render the porosity about 32.5%. Each membrane has an area of 0.825 × 0.825 mm, and a pore array was patterned in an active area of 0.7 × 0.7 mm. Therefore, as 130 membranes were used, the total active area was 63.7 mm^2^. For membranes with 5-μm pores (UTM-5), the total active area was slightly less. Table 1 lists the design values of the UTM filters by the pore sizes and the values measured after fabrication. The measured values were slightly smaller than the design values because the photomask pattern is not accurately transferred to the wafer, given the loading effect that occurs during the fabrication process [40]. Due to the typical negative minus stress applied to the Si_3_N_4_ membrane, the probability of membrane breakage in the process was very low at 0.61%, and the broken membrane was sealed post-treatment. Figure 2b shows a photograph of a UTM filter installed in the filter housing. To eliminate leakage, silicon gaskets were appressed on both sides of the UTM. Figure 2c shows a photograph of the pump head with three input/output ports on the top and 3 × 3 nozzles on the bottom. As shown in the scanning electron micrographs (SEMs) of Figure 2d, the pore side walls are vertical and the diameters are rather uniform. The standard deviations of the pore diameters are the error values of Table 1. The membrane aspect ratio, thus the diameter divided by the thickness (*d*_pore_/*t*_pore_), of UTM-5 is 5.4 but increases to 10.1 for UTM-9. Due to the high aspect ratio, damage applied to cells during filtering can be minimized. Such silicon-based UTM filters are bio-friendly and can be used to prepare transmission electron microscope grids [41]. (More images including a cross-sectional view of the UTM filter are shown in Appendix A).

The simplest methods that evaluate filter cut-offs use beads [11,14]. In Figure 3a, the size distributions of the four types of beads are shown. The average bead sizes were 5.1, 6.8, 8.9, and 10.6 μm and they were mixed in a single suspension. The software uses the outlines to identify particles that are near-circular, calculates their areas and sizes, and fits the sizes to a Gaussian distribution. Figure 3b–e shows the size distributions of beads in the filtrates and retentates when the suspension was filtered through UTM-6, 7, 8, and 9, respectively. Usually, the *D*90 and *D*99 values are used to evaluate size separation performance [42]. *D*99 tends to increase as the pore diameter increases. *D*99 was 6.2 μm for UTM-6, 7.2 μm for UTM-7, 7.9 μm for UTM-8, and 8.8 μm for UTM-9. Major reductions in the retentate peaks are evident, and the second peak at the 6.8-μm position decreases as the pore diameter increases. The *D*99s of all UTM filters were repeatedly evaluated, and the averages and standard deviations are presented in Figure 3f. (Further details on determining the size distribution of beads and microscope images of clogged filters are shown in Appendix A.)

Figure 4 shows the filtering results for K562 cells based on the finite element method. Figure 4a–c shows the filtrate and retentate particle distributions after filtering. A MATLAB code was written to fit the results to a Gaussian distribution. The filtrate peak downshifted as the pore diameter decreased (Figure 4d). The median values (*D*50s) of the filtrate distributions corresponding to the various filter pore diameters are shown in Figure 4e. The *D*50 was 15.7 μm before filtration, and did not change after filtering through UTM-15. However, the values decreased slightly to 15.4 μm after filtering through UTM-9, to 14.7 μm after filtering through UTM-7, and to 13.3 μm after filtering through UTM-5 because the uniform, regular pore arrangements of the UTM filters very effectively fractionate cells [19]; cells can be effectively isolated by controlling pore size. Table 2 and Table 3 list the *D*10, 50, 90, 95, and 99 values of the original suspension, and those of the filtrates and retentates, by the filter pore diameters. The filtrate distribution is greatly affected by the pore diameter, but the retentate distribution is less so. The filter particle recovery rates by the pore diameters are shown in Figure 4f. The particle recovery rate is high in the retentate and low in the filtrate when the pore diameter is small, but high in the filtrate and low in the retentate when the pore diameter is large. However, the total recovery rate is always high, thus 83.6% to 91.3%. (Further results on K562 cell filtration can be found in Appendix A.)

The effect of pore diameter on the pore penetration of K562 cells was investigated via finite element analysis. Figure 4g shows the analytical parameters, thus the cell diameter (*d*_cell_), pore diameter (*d*_pore_), and the pressure difference (Δ*p*) between the upper and lower sides of the membrane. In fact, positive pressure was applied to the upper sides of the membrane and cell. The effect of applied pressure on the cell penetration length (*L*_P_) was calculated for various pore diameters. Figure 4h shows the deformation of cells of diameter 15 μm when passing through membranes with pore diameters of 5, 7, and 9 μm. In all cases, the same pressure difference (140 Pa) was applied. As the pore diameter increases, cell deformation also increases, and therefore *L*_P_. To allow for quantitative analysis, *L*_P_ was written as a function of Δ*p* (Figure 4i). Here, the ‘bonded’ condition assumes that the cell and the UTM are attached, and the ‘frictional’ condition assumes that they slide with a friction coefficient of 0.3. Note that when the pore diameter is 5 μm, the initial *L*_P_ is small and then increases slowly with respect to Δ*p*; when the pore diameter is 9 μm, the initial *L*_P_ is large and increases rapidly with respect to Δ*p*. Thus, as the pore diameter increases, the more readily the cells penetrate the pores, greatly increasing the probability that cells will pass through the filter. The pressure was assumed to be applied to the upper side of the filter, as during experiments. However, a scenario in which suction was applied below the cells was also analyzed; this differed significantly from the pressure-on-top situation, which can be found in Appendix A.

Figure 5a shows the principles of TFF and pumping head filtration (PHF). In Figure 5b, the results of real-time measurement of the filtrate volume per unit area from UTM-7 are shown. When the three filtering methods were compared, TFF and PHF were significantly better than DEF in terms of the amount of filtering at a specified time. DEF proceeded in a non-pressurized condition without a vacuum pump. CML cells are soft, and thus easily damaged by high pressure, undermining the utility of filtering. In the DEF graph, fouling commences within 2 s, thus even under non-pressurized conditions. Based on the measurements and the calculations of Equations (2), (4), (6) and (8), both the *J*_0_ values and the blocking constants can be obtained using the least squares method (LSM). The validities of the blocking models can be compared by determining the extent of *RMSE* minimization; the *RMSE* is:(11)$$RMSE=\sqrt{\frac{\sum_{i=1}^{N}{\left(v_{measured}(i)-v_{calcualted}(i)\right)}^{2}}{N}}$$

Based on the overall experimental results shown in Figure 5b, fitting was performed with four blocking models for the initial 15% of the data. In DEF and TFF, rapid saturation was observed at the beginning, but in PHF, wide linearity was observed without saturation. The DEF result is shown in Figure 5c and is in good agreement with predictions of the intermediate, standard blocking model. Two types of blocking occurred [43]. However, minor differences between the measured *J* values and those calculated using the fitted parameters are apparent in Figure 5d. The minimized *RMSE* values are shown in Table 4. By comparing the minimized RMSE values, the suitability of the four blocking models can be determined. Additionally, the suitability of four blocking models can also be determined by the linear regression fitting method [26]. Therefore, the measured data were fitted by the linear regression models of the blocking laws. The inset of Figure 5c shows the linear regression model for standard blocking; the *R*^2^ value was 0.988. The inset of Figure 5d shows the linear regression model for intermediate blocking; *R*^2^ was 0.989. Figure 5e shows that, during TFF, not only the filtering volume increased but the fouling initiation time was delayed, implying that the filter blocks more slowly even at much higher filtration volumes than that of DEF. Unlike during DEF, flux continues, thus never becoming completely saturated, even after fouling onset [44] (Figure 5a). When the LSM was used to explore the validities of the blocking models, TFF was well-matched to the intermediate blocking model. However, as shown in the inset to Figure 5e, when determining validities using linear regressions, the match to the standard blocking model was also high; *R*^2^ was 0.995. The inset of Figure 5f shows the linear regression result for the intermediate blocking model. Qualitatively, the agreement is good, but *R*^2^ is low because of data instability. 

In the DEF and TFF, curves of the filtrate volume per unit area are nonlinear and become saturated with time (Figure 5c,e), but the curve for PHF is linear and does not become saturated (Figure 5g). When the LSM was used to explore the validities of blocking models, the theoretical curves were in good agreement with the experimental data, but did not indicate a preferred fit for any model. As shown in Figure 5h and inset, the periodic change in the filtering rate is due to the regular backflush in the PHF filtering process. Thus, the filtering results are difficult to explain using the known blocking mechanisms. Table 4 shows that, even after fitting via linear regression, the *R*^2^ values of all four models were less than 0.1, evidencing poor validity. This confirms that PHF cannot be explained by an existing blocking model. Despite this, PHF is superior to DFF and, in the long-term, better than TFF because it maintains a constant flow rate and minimizes fouling by continuously repeating injection and suction. 

Figure 6 shows micrographs of fluorescent K562 cells after filtration. The reason for fluorescent imaging was to observe the K562 cells to determine whether there was damage by filtration. This is an indispensable process for further analysis and application of isolated K562 cells. Figure 6a merges the fluorescent and 4′,6-diamidino-2-phenylindole (DAPI)-stained images of K562 cells. The breakpoint cluster region (BCR) is stained green in Figure 6c. Figure 6d shows the protein-tyrosine protein kinase abl1 gene stained orange. K562 cells are chronic myeloid leukemia (CML) cells. The BCR-ABL gene sequence is created via partial breakage and joining of chromosomes 9 and 22. Thus, BCR-ABL status can be observed using a single filtered cell [45].

## 5. Conclusions

We developed a filtration system based on UTMs with pores of diameter 5 to 9 μm and used this to isolate near-circular cells and beads of various sizes. The UTM thickness was 0.9 μm and the pore diameters were very uniform, enabling accurate size cut-off. UTM filters were applied for the filtering of beads and K562 cells. Whereas the *D*99 of the filtered beads was very close to the pore diameter, the *D*99 of the filtered K562 cells was significantly larger than the pore diameter. This is because beads are rigid and do not deform, but cells are soft and easily deformable, allowing them to pass through pores much smaller than themselves. Therefore, when isolating cells of a specific diameter, a filter with a pore diameter smaller than the cells is required. As the pore diameter became smaller, the size distributions of cells passing through the filter changed from the original value. Specifically, the *D*50 value of K562 cells was 15.7 μm but, after filtering through UTM-5, the figure became 13.2 μm. It was thus possible to determine precisely a cut-off yielding a single, soft cell line.

Apart from the precise size cut-off, the system minimizes fouling and is very reproducible. The reason why fouling is minimized is due to the application of periodic backflush by PHF, but also to the fact that clogged cells can be easily removed due to the ultra-thin thickness and regular pore structure of UTM. Intrinsically, in cases of DEF and TFF, the filtering rate is inevitably reduced when fouling occurs in the filter, but in the case of PHF, the filtering rate is maintained constant because backflush is periodically applied. As such, the feature of preventing fouling and the feature of keeping the filtering rate constant improves the long-term viability of the developed filtering system. Valid blocking models were developed for DEF, TFF, and PHF filtrations. The CSM software facilitates fast and accurate statistical analysis of the sizes of many cells. The system can be applied to separate not only soft cells but also MVs of various sizes. The economic outlook is good given the accurate separation, high reproducibility, and long-term filter viability.

## Figures and Tables

**Figure 1 membranes-13-00707-f001:**
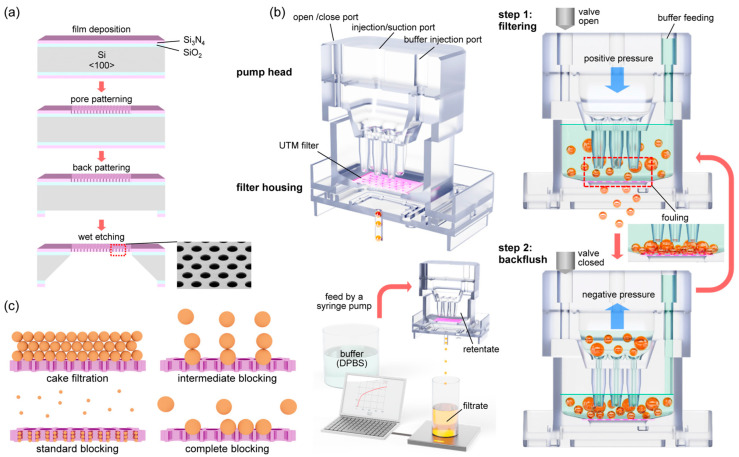
*Overview of the filtering system and filtering principle*: (**a**) Illustrated diagram for the fabrication process of the UTM filter, (**b**) the structure of the PHF (**upper**), mechanism of backflush (**right**), schematic of the filtering system (**lower**), (**c**) illustration of four blocking models: cake filtration, intermediate blocking, standard blocking, and complete blocking.

**Figure 2 membranes-13-00707-f002:**
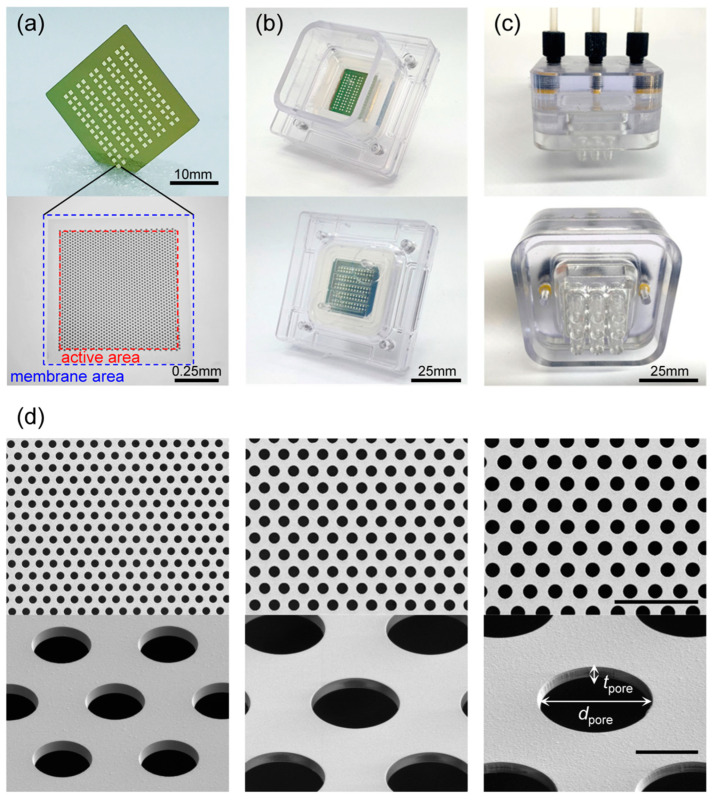
*Structure of UTM filter and PHF*: (**a**) Photo of UTM filter (**upper**), microscope image showing active area and membrane area (**lower**), (**b**) photo of the lower part of PHF (filter housing) with UTM filter installed inside, (**c**) photo of the upper part of PHF (pump head), (**d**) SEM images of UTM filter: UTM-5 μm (**left**), UTM-7 μm (**middle**), and UTM-9μm (**right**) (the scale bar in the upper image represents 50 μm and the scale bar in the lower image represent 5 μm).

**Figure 3 membranes-13-00707-f003:**
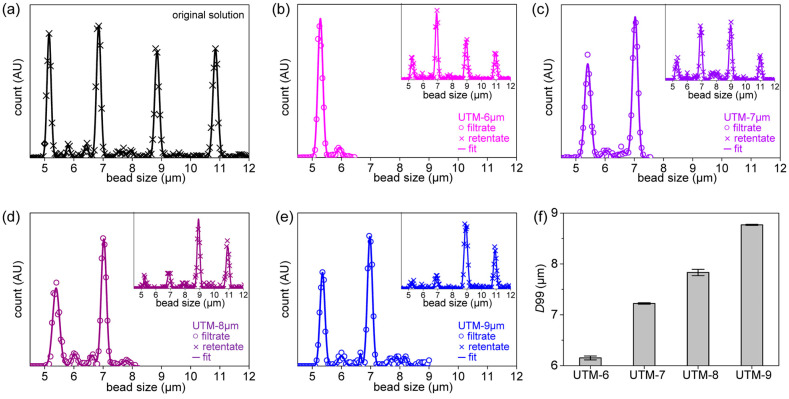
*Filtering result of beads mixture*: (**a**) size distribution of the original beads mixture, (**b**) size distribution of the filtrate after filtration with UTM-6 μm, distribution of retentate (inset), (**c**) distribution of the filtrate of UTM-7 μm, distribution of retentate (inset), (**d**) distribution of the filtrate of UTM-8 μm, distribution of retentate (inset), (**e**) distribution of the filtrate of UTM-9 μm, distribution of retentate (inset), (**f**) *D*99 of filtrate according to UTMs with various pore diameters.

**Figure 4 membranes-13-00707-f004:**
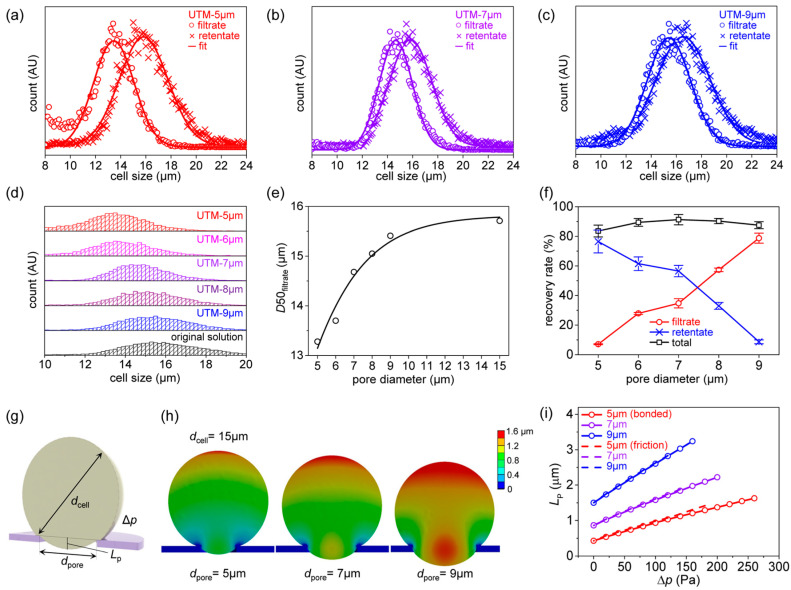
*Filtering result of K562 cell*: (**a**) size distribution of the filtrate and retentate after filtration with UTM-5 μm, (**b**) size distribution of the filtrate and retentate after filtration with UTM-7 μm, (**c**) size distribution of the filtrate and retentate after filtration with UTM-9 μm, (**d**) comparison of size distributions of filtrates from various UTM filters and original solution, (**e**) *D*50 of filtrate as a function of pore diameters, (**f**) recovery rates as a function of pore diameters, (**g**) modeling parameters of cell entering the pore of the membrane, (**h**) finite element analysis (FEA) result of deformation of cell entering the pores of different diameters, (**i**) FEA result of the penetrating length (*L*_P_) as a function of the applied pressure (Δ*p*).

**Figure 5 membranes-13-00707-f005:**
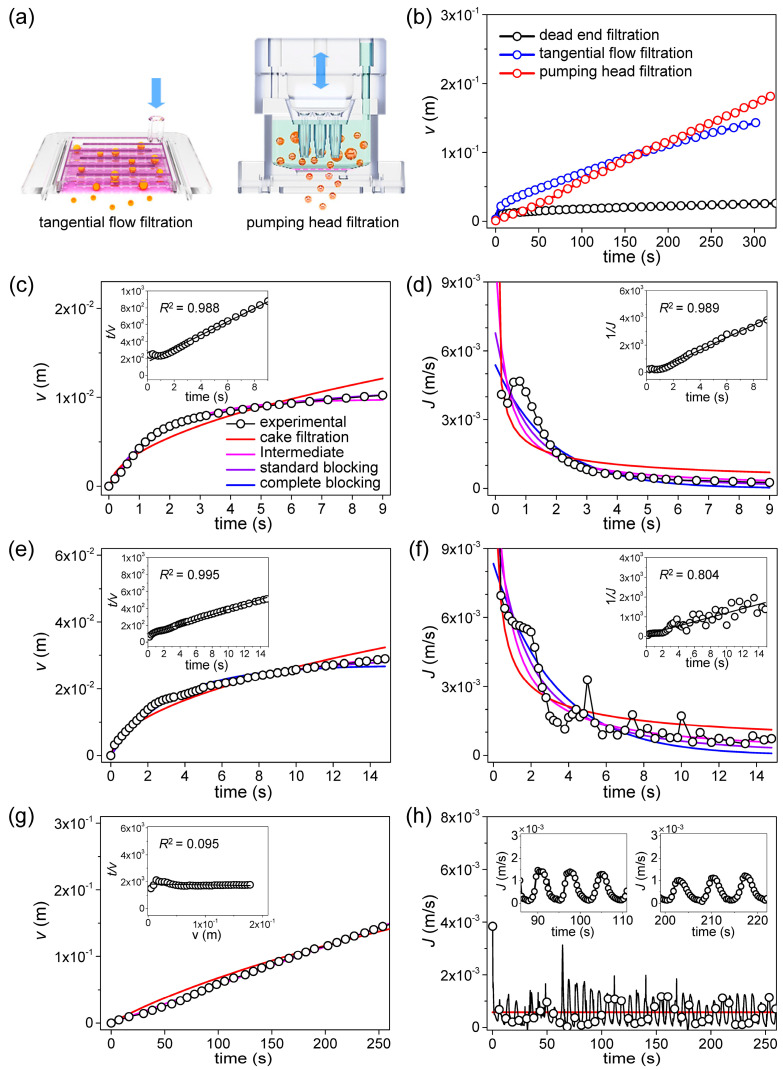
*Analysis of the filtering rate from UTM-7 filtering result*: (**a**) Illustrated principle of two filtering methods: TFF, PHF, (**b**) comparison of filtrate volume per unit membrane area measured from three filtering methods, (**c**) filtrate volume per unit membrane area measured from DEF, calculated curves by four blocking models, the linear regression model for standard blocking (inset), (**d**) filtering rate measured from DEF, calculated curves by four blocking models, the linear regression model for intermediate blocking (inset), (**e**) filtrate volume per unit membrane area measured from TFF, calculated curves by four blocking models, the linear regression model for standard blocking (inset), (**f**) filtering rate measured from TFF, calculated curves by four blocking models, the linear regression model for intermediate blocking (inset), (**g**) filtrate volume per unit membrane area measured from PHF, calculated curves by four blocking models, the linear regression model for cake filtration (inset), (**h**) filtering rate measured from TFF, calculated curves by four blocking models, two graphs for a specific time (inset).

**Figure 6 membranes-13-00707-f006:**
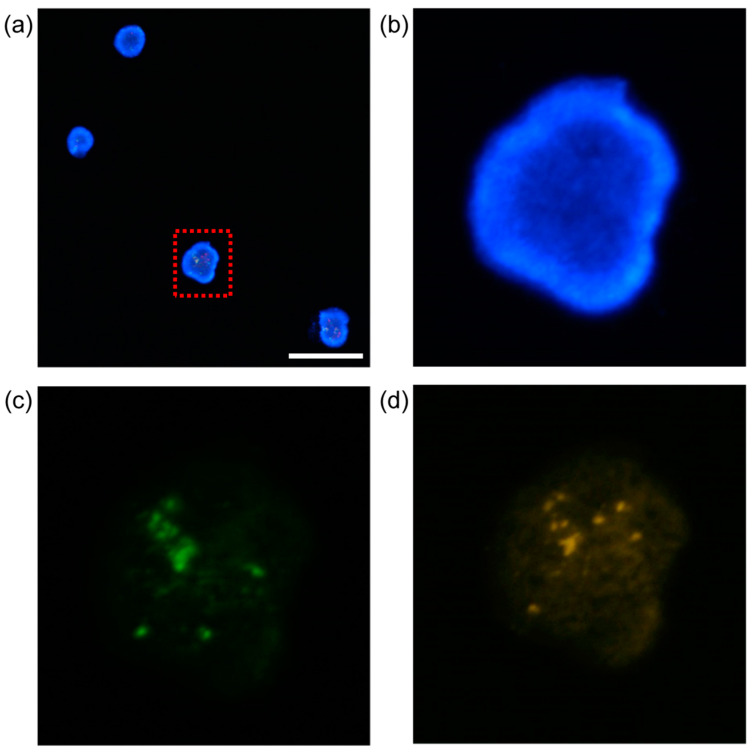
*Florescence microscope image of K562 cell*: (**a**) a merged image of the stained K562 cells, which is stained with 4′,6-diamidino-2-phenylindole (DAPI) by fluorescence microscopy (the scale bar in the image represents 100 μm), (**b**) magnified image of the rectangular part, (**c**) BCR stained in green color, (**d**) ABL stained in orange color.

**Table 1 membranes-13-00707-t001:** Design values of UTM filters according to the pore size and measured values.

	Designed Pore Dia. (μm)	Actual Pore Dia. (μm)	Designed Porosity (%)	Actual Porosity (%)	Active Area (mm^2^)	Thickness (μm)
UTM-5	5	4.82 ± 0.13	32.6	30.3	54.9	0.89
UTM-6	6	5.87 ± 0.01	32.5	31.1	63.7	0.89
UTM-7	7	6.83 ± 0.06	32.5	31.0	63.7	0.89
UTM-8	8	7.60 ± 0.04	32.6	29.6	63.7	0.89
UTM-9	9	9.00 ± 0.06	32.6	32.6	63.7	0.89

**Table 2 membranes-13-00707-t002:** *D*10, 50, 90, 95, and 99 of original solution, and filtrate.

Filtrate	K562	UTM-5	UTM-6	UTM-7	UTM-8	UTM-9	UTM-15
*D*10	13.4	10.0	11.0	13.0	12.7	13.3	13.6
*D*50	15.7	13.3	13.7	14.7	15.1	15.4	15.7
*D*90	18.6	15.4	15.9	16.6	17.4	17.8	18.4
*D*95	19.4	16.1	16.7	17.4	18.1	18.5	19.3
*D*99	22.2	17.5	18.2	18.7	19.7	20.4	22.4

**Table 3 membranes-13-00707-t003:** *D*10, 50, 90, 95, and 99 of original solution, and retentate.

	K562	UTM-5	UTM-6	UTM-7	UTM-8	UTM-9
*D*10	13.4	13.5	13.7	13.8	13.4	13.6
*D*50	15.7	15.8	15.9	15.8	16.0	16.6
*D*90	18.6	18.9	18.8	18.5	18.9	20.0
*D*95	19.4	19.7	19.8	19.5	20.0	21.6
*D*99	22.2	22.4	22.8	23.2	23.6	25.5

**Table 4 membranes-13-00707-t004:** Review of suitability for 4 blocking models for each filtration method for UTM-7 filtering result.

Filtration Method	Blocking Model	Fitting Result			
		*J*_o_ (m/s)	Blocking Constant	Fitting Method 1: *RMSE*	Fitting Method 2: *R*^2^
Dead end filtration	cake filtration	2.46 × 10^−2^	3.24 × 10^5^ (m^−2^s)	2.50 × 10^−3^	0.658
	intermediate	1.26 × 10^−2^	3.67 × 10^2^ (m^−1^)	3.56 × 10^−4^	0.989
	standard	4.82 × 10^−3^	1.35 × 10^2^ (m^−1^)	7.12 × 10^−4^	0.988
	complete	3.26 × 10^−3^	2.45 × 10^−1^ (s^−1^)	1.25 × 10^−3^	0.873
Tangential flow filtration	cake filtration	9.62 × 10^−2^	3.89 × 10^4^ (m^−2^s)	2.68 × 10^−3^	0.893
	intermediate	1.06 × 10^−2^	8.91 × 10^1^ (m^−1^)	1.16 × 10^−3^	0.804
	standard	6.09 × 10^−3^	4.17 × 10^1^ (m^−1^)	2.12 × 10^−3^	0.995
	complete	4.32 × 10^−3^	1.08 × 10^−1^ (s^−1^)	3.06 × 10^−3^	0.736
Pumping head filtration	cake filtration	5.28 × 10^−4^	1.09 × 10^−6^ (m^−2^s)	1.24 × 10^−3^	0.095
	intermediate	5.28 × 10^−4^	1.07 × 10^−10^ (m^−1^)	1.24 × 10^−3^	0.011
	standard	5.74 × 10^−4^	2.00 × 10^−14^ (m^−1^)	1.24 × 10^−3^	0.079
	complete	5.74 × 10^−4^	1.41 × 10^−12^ (s^−1^)	1.24 × 10^−3^	0.028

## Data Availability

Not applicable.

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
