# Peer review of "Precise Filtration of Chronic Myeloid Leukemia Cells by an Ultrathin Microporous Membrane with Backflushing to Minimize Fouling"

_membranes, 2023, doi:10.3390/membranes13080707_

Round 1

Reviewer 1 Report

This study presents a cell filtration platform that affords accurate size separation and minimizes fouling. The results obtained are interesting and merit the publication. For better understanding of readers, it would be desirable to pay attention to the following comments.

 Fig. 5 and Table 5

 Please explain why the initial filtration rate J0 is a fitting parameter.

 Isn’t J0 a constant?

 Why is the time range of the filtration data to be analyzed different?

 DEF: 0-9 s, TFF: 0-14 s, PHF: 0-250 s

Author Response

Response letter for paper entitled "Precise Filtration of Chronic Myeloid Leukemia Cells by an Ultrathin Microporous Membrane with Backflushing to Minimize Fouling" 

<Reviewer 1>

We really appreciated all reviews from the reviewer, it was really helpful for improving our manuscript.    All reviews have been responded and manuscript was updated based on reviews. As the updated parts in the revised manuscript were shown in red color to gain visibility.

Reviewer’s general comments

This study presents a cell filtration platform that affords accurate size separation and minimizes fouling. The results obtained are interesting and merit the publication. For better understanding of readers, it would be desirable to pay attention to the following comments.

→ We sincerely appreciate for your positive evaluation of our efforts on our manuscript. We have sincerely responded to your valuable comments on our manuscript.

Reviewer’s comment

1) Fig. 5 and Table 5

 Please explain why the initial filtration rate J0 is a fitting parameter.

 Isn’t J0 a constant?

→ We appreciate reviewer’s comment on the corresponding part. Yes, J0 is constant. The value was determined by the value at t = 0 in the function by fitting. That's right, the J0 value can be determined by the experimental value, but in the case of the experimental value, the error is large, so it was determined during the fitting process. It is judged that the difference from the actual experimental value is not large.

2) Why is the time range of the filtration data to be analyzed different?

 DEF: 0-9 s, TFF: 0-14 s, PHF: 0-250 s

→ We appreciate reviewer’s comment on the corresponding part. No, it isn't. The analysis was conducted based on the same method. Based on the overall experimental results shown in Fig. 5b, the initial 15% of the data was taken with the program and fitting was performed. However, in DEF, TFF, rapid saturation is observed at the beginning, so the initial range is enlarged and displayed. Since linearity is observed without saturation in PHF over a wide range, the widest range is displayed. Corresponding comments have been added to the manuscript.

Reviewer 2 Report

The manuscript prepared by Lee et al. demonstrated the development of a cell filtration platform, and cell size measurement software. Specifically, the method effectively reduced the size of chronic myeloid leukemia cells and achieved a high recovery rate. The manuscript was written well, the experiment design was valid, and the data presentation was beautiful. I would suggest accpetance of this manuscript after the author to conduct a comprehensive language check.

Minor english revision is needed.

Author Response

Response letter for paper entitled "Precise Filtration of Chronic Myeloid Leukemia Cells by an Ultrathin Microporous Membrane with Backflushing to Minimize Fouling" 

<Reviewer 2>

We really appreciated all reviews from the reviewer, it was really helpful for improving our manuscript.    All reviews have been responded and manuscript was updated based on reviews. As the updated parts in the revised manuscript were shown in blue color to gain visibility.

Reviewer’s general comments

The manuscript prepared by Lee et al. demonstrated the development of a cell filtration platform, and cell size measurement software. Specifically, the method effectively reduced the size of chronic myeloid leukemia cells and achieved a high recovery rate. The manuscript was written well, the experiment design was valid, and the data presentation was beautiful. I would suggest acceptance of this manuscript after the author to conduct a comprehensive language check.

→ We sincerely appreciate for your positive evaluation of our efforts on our manuscript. Although the English of this manuscript was intensively corrected by a professional proofreader, it was carefully checked and corrected once again.

Reviewer’s comment

1) Minor English revision is needed.

→ The English of the manuscript was checked again from beginning to end, and the parts that needed correction were found and corrected. Here are some examples (4 out of 12 corrections in total) of before and after corrections for sentences that needed improvement in English. The actual corrected parts are more than the parts presented below, and the English corrected parts are displayed in blue color. In total, 12 corrections were made. For information, the part marked with red color was changed according to the opinions of other reviewers.

1) At the line 34-35 of the introduction:

Before correction, the sentence was,

‘Such techniques can be applied alone or together.’

After correction, the sentence became, 

'As the method applying antigen-antibody reactions shows high separation efficiency, it has been developed in various ways.'

2) At the line 312-313 of the Results and discussion:

Before correction, the sentence was,

The minimized RMSE values are compared in Table 5; these are consistent with the model. Also, the measured data were fitted using the linear regression models of the blocking laws.

After correction, the sentence became,

By comparing the minimized RMSE values, the suitability of the four blocking models can be determined. Additionally, the suitability of four blocking models can also be determined by the linear regression fitting method.

3) At the line 351-354 of the conclusion:

Before correction, the sentence was,

'Fouling is reduced via periodic PHB; the UPM is extremely thin, and clogged substances are easily removed.'

After correction, the sentence became,

'The reason why fouling is minimized is due to the application of periodic backflush by PHF, but also to the fact that clogged cells can be easily removed due to the ultra-thin thickness and regular pore structure of UTM.'

4) At the line 126-128 of the supporting information:

Before correction, the sentence was,

'In this case, it can be seen that the overall deformation is concentrated in the lower part of the cell. On the other hand, in the condition of pushing from above, it can be seen that the strain is distributed to the top and bottom of the cell.'

After correction, the sentence became,

'Under the suction condition, it can be seen that the overall deformation is rather concentrated in the lower part of the cell. On the other hand, under the pushing condition, it can be seen that the deformation is distributed to the whole part of the cell.'

Reviewer 3 Report

The paper "Precise Filtration of Chronic Myeloid Leukemia Cells by an Ultrathin Microporous Membrane with Backflushing to Minimize Fouling" presents a filtering platform with backflushing for separation of living cells with specific size and provides the reader with a detailed information on filter preparation process, filtration process, and investigation techniques including counting of the particles (cells) in flitrate and retentate, etc.

The paper can be published in "Membranes" journal after some minor points are addressed.

1. Why ultrathin membrane is abbreviated as UPM? Shouldn`t it be UTM? Also, some abbreviations (for example, Pinched flow fractionation (PFF), polycarbonate track etched (PCTE)) are unnecessary since mentioned in text only once. The authorsshould remove these these abbreviations and may also consider further reducing quantity of the abbreviations for better readability.

2. Line 160-161. It is stated that "By applying a deep learning algorithm, 160 dead cells were recognized and excluded." Please add reference to supporting materials so that it will be clear that the algorithm is described there. More references to supporting materials can be added in other places whre appropriate.

Author Response

Response letter for paper entitled "Precise Filtration of Chronic Myeloid Leukemia Cells by an Ultrathin Microporous Membrane with Backflushing to Minimize Fouling" 

<Reviewer 3>

We really appreciated all reviews from the reviewer, it was really helpful for improving our manuscript.    All reviews have been responded and manuscript was updated based on reviews. As the updated parts in the revised manuscript were shown in red color to gain visibility.

Reviewer’s general comments

The paper "Precise Filtration of Chronic Myeloid Leukemia Cells by an Ultrathin Microporous Membrane with Backflushing to Minimize Fouling" presents a filtering platform with backflushing for separation of living cells with specific size and provides the reader with a detailed information on filter preparation process, filtration process, and investigation techniques including counting of the particles (cells) in flitrate and retentate, etc.

→ We sincerely appreciate for your positive comments of our efforts on the manuscript. We have sincerely responded to your valuable comments on our manuscript.

Reviewer’s comment

1) Why ultrathin membrane is abbreviated as UPM? Shouldn`t it be UTM? Also, some abbreviations (for example, Pinched flow fractionation (PFF), polycarbonate track etched (PCTE)) are unnecessary since mentioned in text only once. The authors should remove these abbreviations and may also consider further reducing quantity of the abbreviations for better readability.

→ We appreciate reviewer’s comment on the corresponding part. UPM was used as an abbreviation for ultra-thin porous membrane, but was modified to UTM according to the suggestion. Additionally, the manuscript was modified to not use abbreviation for pinched flow fractionation (PFF) and polycarbonate track etched (PCTE), and deterministic lateral displacement (DLD).

2) Line 160-161. It is stated that "By applying a deep learning algorithm, 160 dead cells were recognized and excluded." Please add reference to supporting materials so that it will be clear that the algorithm is described there. More references to supporting materials can be added in other places where appropriate.

→ We appreciate reviewer’s comment on the corresponding sentence. Regarding a deep learning algorithm, we referred to two papers. Information about those paper are as follows.

  1. Krizhevsky, A.; Sutskever, I.; Hinton, G.E. ImageNet Classification with Deep Convolutional Neural Networks. Commun ACM 2017, 60, 84-90, doi:10.1145/3065386.
  2. Lecun, Y.; Bottou, L.; Bengio, Y.; Haffner, P. Gradient-based learning applied to document recognition. P IEEE 1998, 86, 2278-2324, doi:10.1109/5.726791.

Information of these references have been added to the corresponding part of the supporting information.

Reviewer 4 Report

This work develops a novel pumping head for backflushing(PHB) filtration system to minimize fouling behavior in the separation of Chronic Myeloid Leukemia Cells. Three filtration methods including DEF, TFF, and PHB were compared, and ultra-thin porous membranes with different pore diameters were investigated.  The overall scientific approach is thorough, but there are several issues that should be addressed and resolved before publication in Membranes.

1. Section 2.2

How did the authors control the reciprocating motion of those two syringe pumps to repeat steps 1 and 2 in every 5s? Please clarify whether it's manual or automatic processing. if it’s an automated system, the name of the control system/software should be mentioned.

2. Section 2.4

What’s the average size of the K562 cells? It’s necessary to add this information as this separation is based on size-exclusion.

3. Section 3

The information in Table 1 is the same as shown in Equations (3) – (10), I suggest erasing Table 1 to make the paper more concise.

4. Line 272 & Table 2

Why the total active area was slightly less only in UPM-5? If the number of membranes is fixed at 130, when the pore diameter decreases, the active area should increase. But Table 2 shows that the active area in UPM-6 to UPM-9 was the same as 63.7 mm2. Please explain and clarify it.

5. Fig 4i & Fig S10b

What did ‘bonded’ and ‘friction’ mean in this Fig 4i? The legend in Fig S10b is not clear to read. In lines 363-365, the authors didn’t clarify what difference was observed in the pressure-on-top (push) and pressure-from-bottom (suction) modes.

6. Lines 367-368

It mentioned TFF and PHF were better than DEF, in which aspect? What was the experimental condition for DEF here? It’s gravity-driven?

7. Fig 5b, 5c

In the caption, the meaning of v is incorrect. It’s not ‘filtering volume’ but ‘filtrate volume per unit membrane area’.  

8. Lines 418-424

What message did the author want to convey through Fig 6? Besides, the scale bar is missing.

9. Lines 445-446

How did the authors get this conclusion? Only v and J were investigated in Fig 5, but there was no information about the pressure. 

Lines 399

It shouldn’t be Fig.5a here, it seems to be Fig.5f, please double-check it. Also, check line 408.

Line 407

‘FTT filtering volumes’ should be ‘TFF filtering volumes per unit membrane area’.

Author Response

Response letter for paper entitled "Precise Filtration of Chronic Myeloid Leukemia Cells by an Ultrathin Microporous Membrane with Backflushing to Minimize Fouling" 

<Reviewer 4>

We really appreciated all reviews from the reviewer, it was really helpful for improving our manuscript.    All reviews have been responded and manuscript was updated based on reviews. As the updated parts in the revised manuscript were shown in red color to gain visibility.

Reviewer’s general comments

This work develops a novel pumping head for backflushing (PHB) filtration system to minimize fouling behavior in the separation of Chronic Myeloid Leukemia Cells. Three filtration methods including DEF, TFF, and PHB were compared, and ultra-thin porous membranes with different pore diameters were investigated. The overall scientific approach is thorough, but there are several issues that should be addressed and resolved before publication in Membranes.

→ We sincerely appreciate for your positive evaluation of our efforts on our manuscript. We have sincerely responded to your valuable comments on our manuscript.

Reviewer’s comment

1) In Section 2.2, how did the authors control the reciprocating motion of those two syringe pumps to repeat steps 1 and 2 in every 5s? Please clarify whether it's manual or automatic processing. if it’s an automated system, the name of the control system/software should be mentioned.

→ We appreciate reviewer’s comment on the corresponding part. The filtering system consists of an UTM filter, a pump head, a filter housing, and syringe pump (Fusion 100, Chemyx, USA). To control the step motor in the syringe pump, a Linux-based 8-bit MCU embedded system (ATmega 328) was used. A GUI program was developed by Python for driving the embedded system. These sentences were added to the section 2.2.

2) In Section 2.4, what’s the average size of the K562 cells? It’s necessary to add this information as this separation is based on size-exclusion.

→ We appreciate reviewer’s comment on the corresponding part. The average size of K562 cells in the original solution before filtration was 15.9 μm. The sentence was added to the section 2.4.

3) In Section 3, the information in Table 1 is the same as shown in Equations (3) – (10), I suggest erasing Table 1 to make the paper more concise.

→ We appreciate reviewer’s comment on the corresponding part. Table 1 was removed according to the suggestion.

4) In Line 272 & Table 2, why the total active area was slightly less only in UPM-5? If the number of membranes is fixed at 130, when the pore diameter decreases, the active area should increase. But Table 2 shows that the active area in UPM-6 to UPM-9 was the same as 63.7 mm2. Please explain and clarify it.

→ We appreciate reviewer’s comment on the corresponding part. UPM-5 was designed and manufactured first. After confirming that the production was stable, the active area was increased through some adjustments. This is because the wider the active area, the better the amount of filtering. However, although the active area can affect the filtering speed, it was determined that the size distribution after filtering would be mainly affected by the pore diameter. In addition, in the theoretical analysis, since the filtering speed per unit active area is the main parameter, it was determined that a slight increase in the active area would not be a problem.

5) In Fig 4i & Fig S10b, what did ‘bonded’ and ‘friction’ mean in this Fig 4i? The legend in Fig S10b is not clear to read. In lines 363-365, the authors didn’t clarify what difference was observed in the pressure-on-top (push) and pressure-from-bottom (suction) modes.

→ We appreciate reviewer’s comment on the corresponding part. Regarding caption of Fig 4i, that part was explained in the section 2.8 as follows. "The contacts between cells and filters were assumed to be either bonded or frictional. In the latter case, the friction coefficient was assumed to be 0.3." However, for gain readability, it was mentioned again in the result and discussion part as follows. Here, the 'bonded' condition assumes that the cell and the UTM are attached, and the 'frictional' condition assumes that they slide with the friction coefficient of 0.3. Regarding caption of Fig S10b, we have updated the figure captions by adding more details about the filtering conditions. The updated figure caption is as follows.

Fig. S10. Further finite element analysis (FEA) result about different filtering condition: (a) deformation of cell entering the pores of different diameters, (the upper panel shows the condition of suctioning the cell by applying negative pressure from the bottom of the filter, and the lower panel shows the condition of pushing the cell by applying positive pressure from the top of the filter.) (b) FEA result of the penetrating length (LP) as a function of the applied pressure (Δp). (two filtering conditions, 'push' and 'suction' were compared.)

We also have added sentence about the difference in the pressure-on-top (push) and pressure-from-bottom (suction) modes as follows.

Under the suction condition, it can be seen that the overall deformation is rather concentrated in the lower part of the cell. On the other hand, under the pushing condition, it can be seen that the deformation is distributed to the whole part of the cell.

6) In Lines 367-368, it mentioned TFF and PHF were better than DEF, in which aspect? What was the experimental condition for DEF here? It’s gravity-driven?

→ We appreciate reviewer’s comment on the corresponding part. TFF and PHF were better than DEF in terms of amount of filtering at a specified time. DEF is gravity driven and a photo of DEF condition is shown in the FigS1 of the supporting information. Also we have added a sentence in the supporting information. ‘As shown in Fig S1a, in the case of DEF, filtering was performed only by gravitational force without a pump.’

7) In Fig 5b, 5c, in the caption, the meaning of v is incorrect. It’s not ‘filtering volume’ but ‘filtrate volume per unit membrane area’.

→ We really appreciate reviewer’s comment on the corresponding part. Yes, v is filtrate volume per unit membrane area, not filtering volume. We have corrected those words in the figure caption.

8) In Lines 418-424, what message did the author want to convey through Fig 6? Besides, the scale bar is missing.

→ We appreciate reviewer’s comment on the corresponding part. The reason for fluorescent imaging was to observe the K562 cells to determine whether there was damage by filtration. This is an indispensable process for further analysis and application of isolated K562 cells. In addition, we have added the scale bar to the fluorescent image.

9) In Lines 445-446, how did the authors get this conclusion? Only v and J were investigated in Fig 5, but there was no information about the pressure.

→ We appreciate reviewer’s comment on the corresponding part. We admit that the meaning we are trying to convey in that sentence is misrepresented. Therefore, the sentence in the conclusion has been corrected as follows.

'The reason why fouling is minimized is due to the application of periodic backflush by PHF, but also to the fact that clogged cells can be easily removed due to the ultra-thin thickness and regular pore structure of UTM. Intrinsically, in cases of DEF and TFF, the filtering rate is inevitably reduced when fouling occurs in the filter, but in the case of PHF, the filtering rate is maintained constant because backflush is periodically applied. As such, the feature of preventing fouling and the feature of keeping the filtering rate constant improves the long-term viability of the developed filtering system.'

10) In Lines 399, it shouldn’t be Fig.5a here, it seems to be Fig.5f, please double-check it. Also, check line 408.

→ We really appreciate reviewer’s comment on the corresponding part. There was a mistake in the corresponding sentence. We have corrected the sentence as follows.

‘In the DEF and TFF, curves of the filtrate volume per unit area are nonlinear and become saturated with time (Fig. 5c and e), but the curve for PHF is linear and does not become saturated (Fig. 5g).’

11) In Line 407, FTT filtering volumes’ should be ‘TFF filtering volumes per unit membrane area’.

→ We appreciate reviewer’s comment on the corresponding part. We have already responded in the comment 10.
